# CORE: A Few-Shot Company Relation Classification Dataset for Robust Domain Adaptation.

**Philipp Borchert**[1,2]**, Jochen De Weerdt**[2]**, Kristof Coussement**[1]**,**
**Arno De Caigny**[1]**, Marie-Francine Moens**[3]

[1]IESEG School of Management, Univ. Lille, CNRS, UMR 9221 - LEM -
Lille Economie Management, F-59000 Lille, France
[2]Research Centre for Information Systems Engineering, KU Leuven, Belgium
[3]Department of Computer Science, KU Leuven, Belgium

## Abstract

We introduce CORE, a dataset for few-shot relation classification (RC) focused on company relations and business entities. CORE includes 4,708 instances of 12 relation types with corresponding textual evidence extracted from company Wikipedia pages. Company names and business entities pose a challenge for few-shot RC models due to the rich and diverse information associated with them. For example, a company name may represent the legal entity, products, people, or business divisions depending on the context. Therefore, deriving the relation type between entities is highly dependent on textual context. To evaluate the performance of state-of-the-art RC models on the CORE dataset, we conduct experiments in the few-shot domain adaptation setting. Our results reveal substantial performance gaps, confirming that models trained on different domains struggle to adapt to CORE. Interestingly, we find that models trained on CORE showcase improved out-of-domain performance, which highlights the importance of high-quality data for robust domain adaptation. Specifically, the information richness embedded in business entities allows models to focus on contextual nuances, reducing their reliance on superficial clues such as relation-specific verbs. In addition to the dataset, we provide relevant code snippets to facilitate reproducibility and encourage further research in the field.[1]

## 1 Introduction

Relation classification (RC) is a fundamental task in natural language processing that involves identifying the corresponding relation between a pair of entities based on the surrounding textual context. Several studies highlight the need for more realistic dataset configurations that include None-of-the-Above (NOTA) detection (Gao et al., 2019; Sabo et al., 2021), few-shot evaluation (Baldini Soares

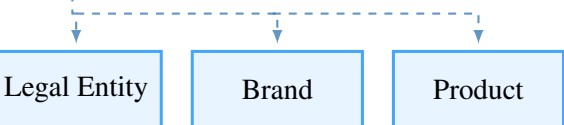

Coke's E1 advertising is pervasive, as one of Woodruff's stated goals was to ensure that everyone on Earth drank Coca-Cola as their preferred beverage.

Legal Entity — Brand — Product

Figure 1: Example for information richness embedded in named business entities.

et al., 2019; Sabo et al., 2021), and out-of-domain adaptation (Gao et al., 2019). However, most existing RC datasets comprise a mixture of domains, and the sets of entities across different domains are often easily distinguishable from one another. The main challenge with these general domain datasets is that few-shot models may take shortcuts during training, given the limited set of possible relation types associated with specific entities (Liu et al., 2014; Lyu and Chen, 2021). For instance, the relation type "place of birth" requires a pair of entities consisting of entity types "Person" and "Geolocation". This can lead to poor generalization performance when evaluated on out-of-domain data.

To address these challenges, we introduce CORE, a high-quality dataset that focuses specifically on company relations. By narrowing the scope to company relations and business entities, CORE presents a more difficult task for models, as business entities embody a variety of information and may be used interchangeably to represent the legal entity, products or services, and brands, depending on the context (see Figure 1). As a result, inferring the relation type solely based on the entities becomes more challenging. Additionally, business entities are well-suited to minimize entity type related clues due to the richness of associated information they possess. By minimizing these clues, CORE aims to incentivize models to more

---

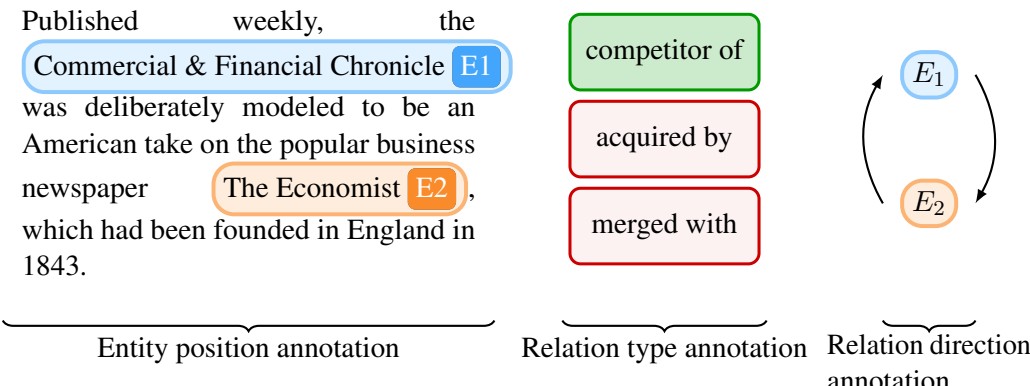

Figure 2: CORE example for an undirected business relation.

effectively leverage textual context and improve predictive performance on out-of-domain data.

The few-shot settings commonly used in distantly supervised datasets are built upon unrealistic assumptions (Sabo et al., 2021). These assumptions include unrealistic relation types, NOTA and entity distributions, as well as a narrow focus on named entities while disregarding common nouns and pronouns. The study shows that models trained on more realistic few-shot evaluation settings exhibit a significant decline in predictive performance. In contrast, CORE overcomes these limitations as it is a human-annotated dataset, not bound by the relations or named entity types present in knowledge bases. It encompasses relation types that are absent in knowledge bases, and includes entity types beyond named entities, such as common nouns and pronouns. On the contrary, distant supervision relies on the extraction of textual context based on subject-relation-object triples in knowledge bases (Han et al., 2018), which restricts the scope to named entities (Sabo et al., 2021).

We contribute to the field of relation classification in three significant ways through our study. First, we introduce CORE, a high-quality dataset that encompasses company relations and extends domain coverage in RC. By focusing on company names and business entities, CORE creates a more demanding benchmark for relation classification models. These entities embody diverse information and can be used interchangeably to represent legal entities, products or services, and brands, depending on the context. Secondly, our benchmark of state-of-the-art models on CORE uncovers substantial performance gaps in few-shot domain adaptation. RC models trained on different domains struggle to adapt to CORE, whereas models trained

on CORE itself outperform other domain-adapted models. Lastly, our study highlights the critical role of high-quality datasets in enabling robust domain adaptation for few-shot RC. Models evaluated on out-of-domain data commonly rely on heuristic patterns and face challenges in adapting to contextual information. In summary, this paper makes a significant contribution by introducing CORE, a novel and high-quality dataset for relation classification. In the context of few-shot domain adaptation, our study demonstrates the challenges faced by state-of-the-art models when dealing with CORE, while highlighting the remarkable performance of models trained on CORE itself.

## 2 Related work

Recent research on **datasets** for few-shot RC has been focused on improving data quality. Several approaches have been proposed, including extensions and augmentations of existing datasets to incorporate features such as NOTA detection or domain adaptation, as well as correction of labeling errors (Alt et al., 2020; Gao et al., 2019).

In this section, we present an overview of recent RC datasets that encompass sentence-level textual context. One of such datasets is TACRED, which is a human-annotated dataset based on online news, focusing on people and organizations as entities (Zhang et al., 2017). It preserves a realistic relation type distribution and includes 79.5% NOTA instances. Alt et al. (2020) re-evaluated the original dataset, analyzing labeling errors, and investigating sources for incorrect model predictions. Another prominent dataset is FewRel, introduced by Han et al. (2018), which consists of text passages extracted from English Wikipedia pages. It is annotated using distant supervision and the Wikidata

knowledge base. The model evaluation is primarily focused on the K-Way N-Shot setting. Gao et al. (2019) extended the dataset by incorporating NOTA examples and introducing out-of-domain test sets. The CrossRE dataset specifically targets the domain adaptation setting in relation classification (Bassignana and Plank, 2022). It incorporates text from Wikipedia and news sources, covering six domains. Notably, the dataset excludes the business domain and is limited to named entities. In terms of relation types, CrossRE adopts a shared approach across domains, omitting any domain-specific relation types. Khaldi et al. (2022) acknowledge the value of business relations extracted from multilingual web sources and present their dataset, which is collected using distant supervision. This dataset focuses primarily on named entities and subsequent human annotation. It contains five types of undirected business relationships.

**Domain adaptation** refers to the task of transferring information from a source domain to a specific target domain. Domain adaptation models learn to extract domain-invariant information from the source domain, which can be applied to the target domain. Popular approaches include domain adversarial networks (Han et al., 2021a; Gao et al., 2019) and leveraging external information such as task descriptions or knowledge bases (Zhang et al., 2021; Zhenzhen et al., 2022). Recent work has also explored prompt tuning approaches (Gao et al., 2021; Schick and Schütze, 2021; Wu and Shi, 2022). In this study, we address the problem of domain adaptation by exposing the model to a limited number of instances from the target domain using few-shot learning. To achieve effective domain adaptation, models combine domain-invariant information learned from the source domain with the information available from the target domain.

## 3 CORE

### 3.1 Data collection

To gather corporate Wikipedia pages in English, we conducted a query on DBpedia (Lehmann et al., 2015) using the ontology class "dbo:Company", which corresponds to the Wikidata identifier "Q4830453". The dataset was constructed using Wikipedia content corresponding to April 2022.

We extract candidate entity pairs based on the company corresponding to the Wikipedia page and named entities in the text. We use a named entity recognition (NER) model based on RoBERTa-Large (Liu et al., 2019) that was fine-tuned on the CoNLL2003 dataset to retain entities tagged as organizations or miscellaneous. The model achieved state-of-the-art performance on CoNLL2003 with a F1-Score of 97% on the full dataset and 96% on entities labeled as organizations. This extraction procedure allows the company corresponding to the Wikipedia page to take any entity type, including named entities, pronouns, and nouns. We use spaCy to create a dictionary of nouns and pronouns in the dataset and manually validate entity types (Honnibal and Montani, 2017). For each candidate entity pair, we extract the corresponding paragraph as supporting context characterizing the relation between the entities. We remove duplicate entity pairs with overlapping context to reduce the number of candidate samples and manually filter out miscellaneous entities that do not belong to the category of business entities, such as "German" or "Dutch". Finally, we randomly sampled 14,000 candidate samples for subsequent annotation.

The annotation platform is created by the authors, offering a similar interface as commercial services such as MTurk (see Appendix A.5). The dataset is annotated by 50 students in business and management who are presented with candidate entity pairs and the corresponding textual context. The annotation consists of selecting one of the predefined relations (see Appendix A.1) and the textual context relevant to determine the selected relation (see Figure 2). Since relations such as "acquired by" imply a direction, annotators are also asked to indicate whether the entity order is reversed: $(e_1, r, e_2) \rightarrow (e_2, r, e_1)$. The relation annotation and the position of $e1$ and $e2$ in the textual context are validated in the subsequent annotation step. Thereby, the validation process includes the removal of candidate instances that contain non-business entities, incorrect relation type selection, insufficient context, or inappropriate data.

The annotators were recruited through an open application process. For their contribution to the annotation process, the students received a reimbursement of $11 per hour. The total compensation for data annotation amounted to approximately $6,600.

### 3.2 Quality control

Data quality is particularly important in low resource settings with models exposed to only a few supporting instances. Alt et al. (2020) find that

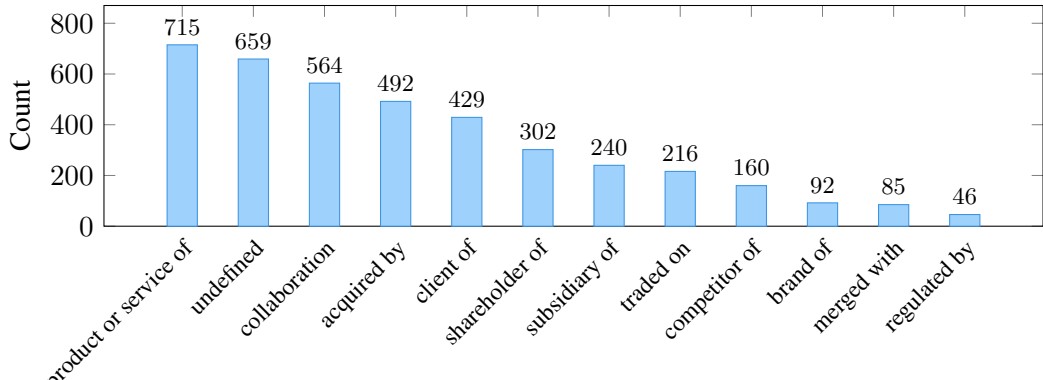

Figure 3: Relation types and distribution in the training set.

incorrectly labeled instances substantially reduce predictive performance in RC models. In close consideration of recent literature, we maintain high data quality in CORE (Alt et al., 2020; Gao et al., 2019; Sabo et al., 2021). The annotation process involves at least three different annotators, with the annotator agreement computed based on relation type annotation, relation direction annotation, and entity annotation. The corresponding Fleiss Kappa is 0.636. We only retain instances with at least 66% inter-annotator agreement.

We educate annotators by providing a mandatory step-by-step tutorial along with guiding documentation on the platform. Further quality control measures include a control set annotated by the authors. Consequently, we excluded annotations from 6 annotators who did not pass the quality control check. We employ consistency checks during the annotation dialog on the platform. For each example, the annotators select business entity types, such as "Company" or "Brand", corresponding to $e1$ and $e2$. The platform validates corresponding relations and entity types, i.e., annotators are required to indicate one of the entities as "product or service" when selecting the "product or service of" relation.

### 3.3 Dataset properties and analysis

The dataset's key properties are presented in Table 1 and discussed in the subsequent section. CORE consists of 12 annotated relation types that were predefined by the authors, as shown in Figure 3. Among these, there is an "undefined" relation that signifies a relationship between $e1$ and $e2$ that does not match any predefined relation type. Thus, "undefined" corresponds to the NOTA category. However, examples containing non-business entities or missing entities were excluded from the dataset.

To better understand the distinctions between CORE and FewRel, which are both based on Wikipedia, we examined their similarities in terms

| Category | | # Train | # Test |
|---|---|---|---|
| Instances | | 4,000 | 708 |
| Relations | Directed | 8 | 8 |
| | Undirected | 4 | 4 |
| | Total | 12 | 12 |
| Entity types | Named entity | 6,777 | 1,214 |
| | Noun | 721 | 114 |
| | Pronoun | 502 | 88 |
| Avg. tokens | | 34 | 34 |

Table 1: Dataset characteristics

of shared instances and relation types. We utilized cosine similarity based on tf-idf values to quantify the textual similarity between instances. Subsequently, we manually analyzed 76 instance pairs that exhibited a similarity greater than 50%. Our analysis revealed nine pairs with matching context, among which one instance pair shared entities and relation type annotations. The remaining pairs focused on different entities and served as examples for various relation types. Examples of these instances are provided in Table 2.

Furthermore, we performed an analysis of the Wikidata properties used in FewRel, including P449 (original broadcaster), P123 (publisher), P178 (developer), P176 (manufacturer) and P750 (distributed by). We observed that all of these properties align with CORE's "product or service of" relation type. Furthermore, we found that the properties P355 (has subsidiary) and P127 (owned by) partially overlap with "subsidiary of" and "acquired by" in CORE. However, these properties include non-business entities and have a broader interpretation of "subsidiary" and "ownership", encompassing concepts like "part-of", physical location, or control. Examples illustrating this overlap are shown in Appendix A.2.

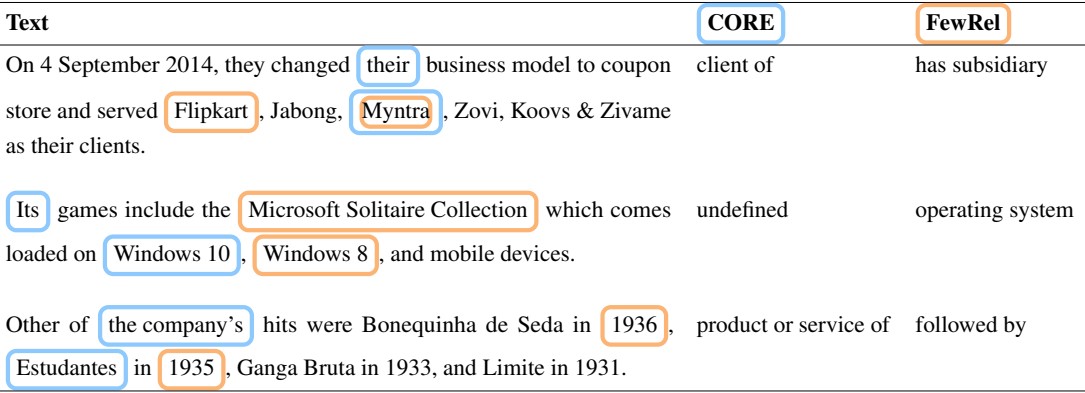

| Text | CORE | FewRel |
|------|------|--------|
| On 4 September 2014, they changed their business model to coupon store and served Flipkart, Jabong, Myntra, Zovi, Koovs & Zivame as their clients. | client of | has subsidiary |
| Its games include the Microsoft Solitaire Collection which comes loaded on Windows 10, Windows 8, and mobile devices. | undefined | operating system |
| Other of the company's hits were Bonequinha de Seda in 1936, Estudantes in 1935, Ganga Bruta in 1933, and Limite in 1931. | product or service of | followed by |

Table 2: Shared instances between CORE and FewRel with corresponding entities highlighted in blue and orange, respectively.

### 3.4 Dataset splits

The annotated instances were randomly divided into a training set (4,000 instances) and a test set (708 instances), each split containing all available relation types. We do not include a dedicated development set, but use part of the training set for model development (Mukherjee et al., 2021).

## 4 Experiments

### 4.1 Task formulation

We adopt the N-way K-shot evaluation setting, where the model is exposed to a limited number of instances $N \times K$ during training. The episodes sampled from the dataset consist of two parts: (1) a support set $S = \{(x_i, r_i)\}_{i=1}^{K \times N}$ containing input $x$ along with $N$ randomly sampled relations $r \in \mathcal{R}$ as well as $K$ instances for each relation and (2) a query set $Q = \{x_i\}_{i=1}^{K \times N}$ containing $K$ instances of the $N$ relations not included in the support set (Gao et al., 2019, 2021).

In line with recent literature, we emphasize the importance of NOTA instances in the few-shot evaluation setting (Gao et al., 2019). By including NOTA as an additional relation type, we effectively sample $N + 1$ relations in each episode. Thereby, the NOTA relation consists of the "undefined" relation type and any relations not used for the training and test episodes (Sabo et al., 2021).

### 4.2 Episode sampling

The sampling procedure accommodates overlapping relation types in the dedicated training and test sets. We consider two disjoint sets of relation types $r_{\text{train}} \cap r_{\text{test}} = \varnothing$ for training and test episodes, respectively. We uniformly sample $N \times K$ support instances from the corresponding set of relation types. This exposes models to the same number of support instances per relation, disregarding the relation distribution in the dataset. Considering the relation distribution in the model evaluation, we sample $N \times K$ query instances randomly (Sabo et al., 2021). All instances included in training episodes are sampled from the training set. For test episodes we sample support instances from the training set and query instances from the test set.

### 4.3 Models

To facilitate the comparison of evaluation results, we benchmark SOTA models in few-shot RC literature, considering performance and prior application in the domain adaptation setting (Sabo et al., 2021; Gao et al., 2021). We utilize BERT-Base with 110 million parameters (Devlin et al., 2019) as an encoder for all benchmarked models. This prevents information disparities between language models, which are commonly trained on Wikipedia data.

**Prototypical Networks (Proto)** create relation type prototypes using the support examples (Snell et al., 2017). Query instances are matched to relation prototypes by computing similarity scores.

In the **BERT-Pair** approach, query instances are concatenated with support instances computing a similarity probability to determine whether the query instances share the same relation type as the support instances (Gao et al., 2019).

In the **BERT with Entity Markers (BERT$_{\text{EM}}$)** approach, the entities are embedded with entity marker tokens at the respective start and end positions in the context (Baldini Soares et al., 2019). The concatenated representation of the entity start markers is used as the example representation. Query instances are classified based on similarity to the support instances.

For the **BERT-Prompt** approach, we express

the relation classification task as a masked language modeling (MLM) problem (Gao et al., 2021; Schick and Schütze, 2021). Given a template $\mathcal{T}$, each input is converted into $x_{\text{prompt}} = \mathcal{T}(x)$ containing at least one [MASK] token. Similarly to the MLM task performed during pretraining of a language model $\mathcal{M}$, the masked token is predicted based on the surrounding context. To accommodate differences in model vocabularies, the relations are mapped to individual tokens included in the vocabulary: $\mathcal{R} \mapsto \mathcal{V}_{\mathcal{M}}$. Given input $x$, the probability of predicting relation $r$ is denoted as:

$$
\begin{aligned}
p(r \mid x) &= p([\text{MASK}] = w_v \mid x_{\text{prompt}}) \\
&= \frac{\exp\left(w_v \cdot h_{[\text{MASK}]}\right)}{\sum_{v' \in \mathcal{V}} \exp\left(w_{v'} \cdot h_{[\text{MASK}]}\right)},
\end{aligned} \quad (1)
$$

where $h_{[\text{MASK}]}$ is the hidden vector of [MASK] and $w_v$ denotes the pre-softmax vector corresponding to $v \in \mathcal{V}$ (Gao et al., 2021; Han et al., 2022).

The **Hybrid Contrastive Relation-Prototype (HCRP)** approach is designed to learn from relation types that are challenging to distinguish. It employs a hybrid approach, creating prototypes and integrating additional data in the form of relation descriptions. HCRP incorporates a task-adaptive focal loss and utilizes contrastive learning techniques to enhance the discriminative power of the prototypes (Han et al., 2021b).

### 4.4 Experimental setup

We evaluate RC models in the few-shot domain adaptation setting for CORE, FewRel and TACRED. Our evaluation process consists of two steps. First, we evaluate the in-domain model performance. In the second step, we evaluate the out-of-domain performance on the respective test sets. We keep the number of training episodes consistent across datasets to prevent information disparity across domains. For each dataset, we sampled 22,500 training episodes, which corresponds to approximately 10 epochs for $K = 1$ on CORE. In the case of FewRel, which has an artificial relation type distribution, we followed the approach described in (Gao et al., 2019). They uniformly sample support and query instances from relation types, which also obscures the distribution of NOTA instances in the underlying data source. To address this, we selected a 50% NOTA rate (Gao et al., 2019) and applied the same sampling procedure to all models trained on FewRel, as in (Sabo et al., 2021).

| Domain | Model | 5-1 | 5-5 |
|---|---|---|---|
| In-domain | Proto-BERT | 26.54 / 26.92 | 31.96 / 43.87 |
| | BERT-Pair | 26.77 / 25.06 | 34.78 / 34.18 |
| | BERT$_{\text{EM}}$ | 36.27 / 44.82 | 44.53 / 56.04 |
| | BERT-Prompt | **53.27 / 67.30** | **74.16 / 88.07** |
| | HCRP | 34.82 / 38.69 | 47.40 / 50.15 |
| FewRel | Proto-BERT | 19.46 / 21.92 | 16.71 / 24.55 |
| | BERT-Pair | 18.74 / 18.86 | 16.58 / 22.79 |
| | BERT$_{\text{EM}}$ | 26.65 / 33.97 | 30.44 / 34.22 |
| | BERT-Prompt | 25.01 / 36.35 | 33.26 / 35.70 |
| | HCRP | 23.54 / 37.49 | 25.41 / 41.67 |
| TACRED | Proto-BERT | 17.93 / 18.66 | 15.13 / 21.14 |
| | BERT-Pair | 17.50 / 17.00 | 16.77 / 16.95 |
| | BERT$_{\text{EM}}$ | 26.47 / 30.38 | 27.38 / 28.79 |
| | BERT-Prompt | 29.61 / 44.12 | 37.88 / 42.60 |
| | HCRP | 27.23 / 32.85 | 41.91 / 41.61 |

Table 3: Micro F1 / Macro F1 for in-domain and out-of-domain models evaluated on CORE.

## 5 Results and discussion

We evaluate the performance of the models using two evaluation metrics: micro F1 and macro F1. Micro F1 provides an overall assessment of model performance by treating instances equally and accounting for the impact of an imbalanced NOTA ratio. In contrast, macro F1 focuses on individual class performance, treating each class equally and remaining unaffected by the NOTA ratio. We report micro F1 and macro F1 scores in percentage format in Tables 3, 4, and 5. The best results in each table are highlighted in bold, while the best performance measures for each domain are underlined. To ensure robust evaluation, the models were evaluated on ten sets of training and test relations for each dataset and N-Way K-Shot setting. The resulting evaluation metrics were then averaged.

### 5.1 Results CORE

The in-domain results presented in Table 3 illustrate the superior performance of the BERT-Prompt model across all few-shot settings, surpassing other approaches in terms of both micro F1 and macro F1 scores. Additionally, our findings reveal that the BERT$_{\text{EM}}$ model exhibits enhanced performance compared to the Proto-BERT and BERT-Pair models, as indicated by significant improvements in both micro F1 and macro F1 metrics.

### 5.2 Results domain adaptation

Out-of-domain evaluation tests models' ability to transfer information across different domains. Following the approach in Gao et al. (2019), models learn to establish mappings between support and

query samples using the source domain data and corresponding labels ($r_{\text{train}}$). In-domain evaluation is performed on $r_{\text{test}}$, where even within this context, label spaces do not overlap. To accomplish this, the model is exposed to $N \times K$ support examples from the target domain. Out-of-domain evaluation follows the same procedure, but with $r_{\text{test}}$ and support samples from a different domain.

Table 3 presents the performance of models trained on FewRel and TACRED, evaluated on CORE. These results reveal substantial performance gaps between out-of-domain models and their in-domain counterparts, underscoring the challenges of generalizing from FewRel and TACRED to CORE. Notably, BERT-Prompt trained on TACRED, achieves the highest performance among the evaluated models, though it still falls significantly short of the performance of the in-domain models. Interestingly, the performance differences among the other out-of-domain models are minimal, implying that all modeling approaches struggle to generalize to CORE. Furthermore, Proto-BERT, BERT-Pair, and BERT$_{\text{EM}}$ models trained on TACRED consistently demonstrate lower macro F1 values compared to the models trained on FewRel.

To investigate the performance of in-domain models on out-of-domain datasets, we compared them with out-of-domain results for FewRel and TACRED, as presented in Tables 4 and 5. Our findings show that the Proto-BERT, BERT$_{\text{EM}}$ and HCRP models trained on CORE consistently outperform the in-domain models trained on FewRel, while the in-domain BERT-Prompt model remains the best performing model. The best performing out-of-domain models are HCRP trained on CORE and HCRP trained on TACRED.

It is worth noting that both CORE and FewRel are based on examples extracted from Wikipedia. While lexical similarity can partially account for the performance differences between the models trained on CORE and TACRED in Table 4, models trained on FewRel fail to outperform the models trained on TACRED, as shown in Table 3.

Regarding domain adaptation results on the TACRED dataset, we observe that the micro F1 performance of the BERT-Prompt model trained on the CORE dataset is comparable to that of the in-domain model trained on TACRED. However, the macro F1 scores of the out-of-domain models are substantially lower. Furthermore, it is evident that Proto, BERT-Pair, and BERT$_{\text{EM}}$ models, includ-

| Domain | Model | 5-1 | 5-5 |
|--------|-------|-----|-----|
| In-domain | Proto-BERT | 20.48 / 25.40 | 19.86 / 20.75 |
| | BERT-Pair | 21.66 / 21.32 | 23.87 / 33.78 |
| | BERT$_{\text{EM}}$ | 35.25 / 48.04 | 46.43 / 51.50 |
| | BERT-Prompt | **56.02 / 71.71** | 59.80 / **86.81** |
| | HCRP | 36.33 / 51.07 | 52.09 / 63.11 |
| CORE | Proto-BERT | 26.19 / 31.64 | 29.82 / 39.52 |
| | BERT-Pair | 18.55 / 20.35 | 18.99 / 22.98 |
| | BERT$_{\text{EM}}$ | 40.27 / 52.42 | 54.03 / 65.45 |
| | BERT-Prompt | 45.08 / 48.01 | 51.82 / 39.05 |
| | HCRP | 51.56 / 57.55 | 61.24 / 67.96 |
| TACRED | Proto-BERT | 20.28 / 23.53 | 20.95 / 27.62 |
| | BERT-Pair | 16.44 / 16.66 | 13.78 / 17.67 |
| | BERT$_{\text{EM}}$ | 34.19 / 43.95 | 38.88 / 47.29 |
| | BERT-Prompt | 48.18 / 54.80 | 53.82 / 50.50 |
| | HCRP | 49.15 / 53.89 | **61.99** / 63.33 |

Table 4: Micro F1 / Macro F1 for in-domain and out-of-domain models evaluated on FewRel. The models are evaluated on the development set corresponding to the Wikipedia data.

| Domain | Model | 5-1 | 5-5 |
|--------|-------|-----|-----|
| In-domain | Proto-BERT | 28.50 / 14.68 | 25.33 / 20.38 |
| | BERT-Pair | 24.25 / 17.71 | 30.57 / 18.72 |
| | BERT$_{\text{EM}}$ | 25.19 / 19.16 | 44.13 / 60.39 |
| | BERT-Prompt | 35.70 / **63.00** | **67.81** / **74.02** |
| | HCRP | 29.58 / 26.43 | 57.28 / 35.61 |
| CORE | Proto-BERT | 15.72 / 24.67 | 11.26 / 29.20 |
| | BERT-Pair | 14.47 / 17.48 | 11.44 / 21.89 |
| | BERT$_{\text{EM}}$ | 16.32 / 38.65 | 18.80 / 48.39 |
| | BERT-Prompt | **36.08** / 45.07 | 65.46 / 35.53 |
| | HCRP | 34.04 / 37.54 | 47.20 / 44.76 |
| FewRel | Proto-BERT | 16.36 / 22.29 | 10.74 / 26.70 |
| | BERT-Pair | 13.16 / 18.47 | 8.75 / 19.69 |
| | BERT$_{\text{EM}}$ | 12.81 / 40.07 | 15.77 / 44.51 |
| | BERT-Prompt | 18.71 / 49.10 | 38.21 / 45.67 |
| | HCRP | 13.72 / 33.76 | 33.17 / 40.03 |

Table 5: Micro F1 / Macro F1 for in-domain and out-of-domain models evaluated on TACRED.

ing those trained on FewRel, struggle to adapt to TACRED when considering the micro F1 results. When examining the macro F1 scores, we find that the BERT$_{\text{EM}}$ and BERT-Prompt models trained on FewRel demonstrate relatively better performance compared to the models trained on CORE. However, this improvement is not reflected in the micro F1 scores.

To demonstrate our model's performance on domain-specific datasets, as opposed to general domain datasets, we included PubMed (Gao et al., 2019) and SCIERC (Luan et al., 2018) in our evaluation. PubMed contains examples extracted from biomedical literature, featuring 10 unique relation types. Notably, both FewRel and PubMed are authored by Gao et al. (2019) and share certain characteristics, including artificially balanced relation

| Domain | Model | 5-1 | 5-5 |
|--------|-------|-----|-----|
| CORE | Proto-BERT | 22.25 / 21.38 | 27.22 / 28.48 |
|  | BERT-Pair | 17.79 / 19.31 | 15.24 / 19.10 |
|  | BERT$_{EM}$ | 38.87 / 43.84 | **56.13** / 62.50 |
|  | BERT-Prompt | 24.68 / 25.18 | 35.85 / 41.09 |
|  | HCRP | 29.74 / **52.35** | 34.96 / 55.66 |
| FewRel | Proto-BERT | 24.63 / 25.14 | 26.36 / 27.43 |
|  | BERT-Pair | 17.39 / 18.91 | 24.53 / 32.00 |
|  | BERT$_{EM}$ | **42.01** / 34.11 | 52.80 / **62.81** |
|  | BERT-Prompt | 25.88 / 41.81 | 28.27 / 41.56 |
|  | HCRP | 32.41 / 40.21 | 37.99 / 36.50 |
| TACRED | Proto-BERT | 19.37 / 20.26 | 19.64 / 21.61 |
|  | BERT-Pair | 15.61 / 16.05 | 13.57 / 16.28 |
|  | BERT$_{EM}$ | 32.12 / 36.99 | 38.72 / 40.32 |
|  | BERT-Prompt | 29.36 / 36.00 | 34.60 / 35.14 |
|  | HCRP | 32.00 / 45.13 | 37.14 / 48.63 |

Table 6: Micro F1 / Macro F1 for out-of-domain models evaluated on PubMed.

| Domain | Model | 5-1 | 5-5 |
|--------|-------|-----|-----|
| CORE | Proto-BERT | 17.71 / 17.78 | 19.37 / 19.83 |
|  | BERT-Pair | 17.69 / 17.54 | 20.42 / 20.03 |
|  | BERT$_{EM}$ | 22.20 / 24.94 | 29.28 / **33.69** |
|  | BERT-Prompt | **41.31** / **28.13** | 47.88 / 26.57 |
|  | HCRP | 18.54 / 22.31 | 18.41 / 23.43 |
| FewRel | Proto-BERT | 16.10 / 17.27 | 16.21 / 18.00 |
|  | BERT-Pair | 16.51 / 17.88 | 17.13 / 19.57 |
|  | BERT$_{EM}$ | 20.41 / 21.63 | 21.78 / 25.57 |
|  | BERT-Prompt | 32.06 / 23.95 | 36.15 / 23.18 |
|  | HCRP | 14.20 / 22.21 | 15.40 / 24.87 |
| TACRED | Proto-BERT | 15.31 / 16.93 | 16.54 / 17.06 |
|  | BERT-Pair | 15.46 / 15.57 | 16.04 / 15.77 |
|  | BERT$_{EM}$ | 17.74 / 20.14 | 17.31 / 20.97 |
|  | BERT-Prompt | 39.59 / 24.67 | **48.08** / 22.56 |
|  | HCRP | 21.03 / 23.25 | 33.66 / 24.98 |

Table 7: Micro F1 / Macro F1 for out-of-domain models evaluated on SCIERC.

distributions. SCIERC comprises examples from scientific literature within the AI domain, including 7 unique relation types. It provides a distinctive set of relation types and follows a different data extraction procedure, adding diversity to our benchmarked datasets. Both of these datasets have a limited number of relation types and do not include an annotated NOTA category. Therefore, following the procedure outlined in Section 4, the inclusion of in-domain models is not feasible.

The out-of-domain evaluation on PubMed, presented in Table 6, demonstrate that, on average, the BERT-EM model trained on CORE outperforms BERT-EM models trained on FewRel and HCRP trained TACRED by 2.40% and 9.61%, respectively. The out-of-domain evaluation results for models evaluated on SCIERC are provided in Table 7. The BERT-Prompt model trained on CORE out-

|  | CORE | FewRel | TACRED |
|--|------|--------|--------|
| CORE | - | 11.75 % | 20.72 % |
| FewRel | 53.41 % | - | 37.90 % |
| TACRED | 44.85 % | 15.12 % | - |

Table 8: Average performance decrease between best performing in-domain models (columns) and out-of-domain models (rows).

performs BERT-Prompt models trained on FewRel and TACRED by 7.14% and 2.25%, respectively.

In summary, our findings suggest that out-of-domain models face challenges in adapting to the CORE dataset. These models consistently exhibit lower performance, both in terms of micro F1 and macro F1 scores. To provide an overview of the performance differences, we illustrate the average performance gaps between the best performing in-domain model and the out-of-domain models across all evaluation metrics in Table 8. Notably, the models trained on CORE show the best average out-of-domain performance, indicated by lower average performance differences in comparison to in-domain models. Furthermore, our experiments show that models trained on CORE outperform those trained on FewRel when evaluated on datasets with a narrow domain focus, such as PubMed and SCIERC. Several factors contribute to these performance differences. First, the information richness in the training data allows the models to focus on contextual nuances, reducing their reliance on superficial clues such as relation-specific verbs. Second, the broader entity coverage, including named entities, nouns, and pronouns, exposes models trained on CORE to more linguistically diverse instances for the different relation types. These observations align with the findings of Mao et al. (2022), who report significant out-of-domain performance improvements in low-resource settings when training on high-quality data. Considering the benchmark evaluation, the prompt tuning approach emerges as the best-performing model.

## 5.3 Error analysis

We conducted a qualitative analysis to identify error patterns in the domain adaptation setting. We used prediction examples from the BERT-Prompt models trained on different domains, which achieved the highest average performance in the previous section. From these models, we randomly sampled 100 erroneous and 100 correct predictions, focusing on the 5-Way 5-Shot evaluation setting.

For models adapted to CORE (as shown in Table 3), we find that correct predictions often included (i) verbs closely related to the target relation, such as "sold", "bought", "merge" or (ii) combinations of entities that revealed the relation type, such as "it is listed on the New York Stock Exchange" or "the company was cited by the Food and Drug Administration". While there were no discernible patterns in the context, the domain-adapted models performed better on NOTA instances compared to instances belonging to the $N$ relations of interest that did not exhibit patterns (i) and (ii). Selected examples are displayed below.

**Erroneous predictions for out-of-domain models evaluated on CORE.**

- *competitor of*: In 2003, Toronto – dominion bank held talks to merge Waterhouse E1 with E*TRADE, which would have created the second - largest discount broker in the united states after Charles Schwab E2 , but the two sides could not come to an agreement over control of the merged entity.

- *product or service of*: On July 4, 2011, Cope-Com E2 had converted their famed and legendary Amiga E1 game called battle squadron to iOS devices titled battle squadron one and published the game through apple app store.

**Correct predictions for out-of-domain models evaluated on CORE.**

- *listed on*: Sopheon E1 is listed on the alternative investment market of the London Stock Exchange E2 .

- *acquired by*: In 2008 the business was restructured with the creation of a new Irish resident holding company, Charter International plc. E1 the company was acquired by Colfax corporation E2 , an American company, in January 2012.

Our analysis revealed that models trained on CORE recognize typical patterns based on verbs and entity types, as well as more complex textual examples that require contextual understanding. Such examples include instances without verbs, instances involving multiple possible relations, or ambiguous entity types, as illustrated in Figure 1.

## 6  Conclusion

In this paper, we introduced CORE, a few-shot RC dataset that specifically focuses on company relations and business entities. To the best of our knowledge, there are no other datasets that specifically target company relations for few-shot RC, making CORE a valuable addition to the existing benchmark resources. By extending the availability of datasets for few-shot domain adaptation settings, CORE contributes towards more robust and realistic RC evaluation results.

Despite the progress made in addressing challenges in RC datasets, such as incorporating more realistic entity types and relation distributions, there is still a need for further research and attention in adapting models to out-of-domain datasets using diverse datasets and modeling approaches. Our findings illustrate that models trained on other RC datasets often rely on shortcuts that fail to generalize to CORE. Moreover, models trained on CORE display superior out-of-domain performance. The information richness in the training data enables models to focus on contextual nuances, reducing their reliance on superficial clues such as relation-specific verbs. This underscores the significance of high-quality datasets that include challenging entity types and require a deeper contextual understanding. In conclusion, CORE serves as a valuable resource for the few-shot domain adaptation setting in RC. It highlights the importance of incorporating diverse and challenging datasets and encourages future research efforts to focus on improving model adaptation to out-of-domain data. By addressing these challenges, we can advance the development of more effective and robust RC models.

## Limitations

CORE instances are solely extracted from company Wikipedia pages, which introduces a limitation in terms of lexical similarity within the contextual information. To address this constraint, diversifying the dataset by incorporating company relations from various data sources, such as business news, would introduce linguistic variety and establish a more rigorous benchmark for evaluation.

The dataset's practical applications are restricted to low-resource settings due to the limited number of instances and relations it contains. Expanding the dataset by including more instances and relationships incurs substantial costs, as it necessitates human annotation efforts. Out of the 14,000 an-

notated candidate instances, only 30% contain any of the predefined relations. However, the current inclusion of the "undefined" category fails to accurately reflect the distribution of invalid candidate instances. This discrepancy primarily arises from the absence of entities within the context or entities that fall outside the scope of business entities.

Another limitation lies in the dataset being monolingual, as it is currently limited to English. The transfer of annotations from the English Wikipedia corpus to other languages would require additional human supervision and allocation of resources.

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

# A Appendix

## A.1 Relations

In the following, we provide textual descriptions of the relations included in the dataset. The relations and their descriptions were predefined by the authors.

- Acquired by: $e2$ purchases controlling stake in $e1$. The relation is directed. The inverted relation is best described by the same relation type.

- Brand of: $e2$ offers products or services of $e1$ (Brand). The relation is directed. The inverted relation is best described by the same relation type.

- Client of: $e1$ uses (and presumably pays for) products or services offered by $e2$. The relation is directed. The inverted relation is best described by "Supplier of".

- Collaboration: $e1$ and $e2$ collaborate in (parts of their) business activities. The relation is undirected.

- Competitor of: $e1$ competes for resources with $e2$. The relation is undirected.

- Merged with: $e1$ and $e2$ merged (parts of) their business activities. The relation is undirected.

- Product or service of: $e1$ is offered for commercial distribution by $e2$. The relation is directed. The inverted relation is best described by the same relation type.

- Regulated by: $e2$ regulates (parts of) the business activity of $e1$. The relation is directed. The inverted relation is best described by the same relation type.

- Shareholder of: $e1$ owns shares in $e2$. The relation is directed. The inverted relation is best described by the same relation type.

- Subsidiary of: $e2$ legally owns $e1$. The relation is directed. The inverted relation is best described by "Parent of".

- Traded on: Shares of $e1$ are listed on $e2$ (Stock exchange). The relation is directed. The inverted relation is best described by "lists".

- undefined: $e1$ and $e2$ are business entities. The relation between the entities cannot be described by any other defined relation. The relation can be directed or undirected, however directed relations are not annotated as such.

## A.2 Dataset Analysis

In contrast to the CORE relation "subsidiary of" and "acquired by", the FewRel relation P355 (has subsidiary) encompasses non-business entities and offers a broader interpretation of the term "subsidiary" beyond the corporate parent or subsidiary relationship.

- *has subsidiary*: The 2017–18 UMass [E1] Minutewomen basketball team represents the University of Massachusetts Amherst [E2] during the 2017–18 college basketball season.

- *has subsidiary*: Davidson County is also served by Davidson County Community College [E2], a comprehensive community college that is a member school of the North Carolina Community College System [E1].

The FewRel relation P127 (owned by) exhibits partial overlap with the CORE relations "subsidiary of" and "acquired by". However, it introduces a different perspective on ownership, encompassing a range of interpretations that extend beyond conventional notions of mere possession, including the notion of being a constituent part of an entity, physical location, and even physical control.

- *owned by*: The school is located in the city of Mannheim, Baden - Württemberg [E2] in Germany at Mannheim Palace [E1], one of the largest baroque castles in Europe.

- *owned by*: After they returned to the Polanski [E2] residence on Cielo Drive [E1], Patricia Krenwinkel, Susan Atkins and Charles "Tex" Watson entered the home.

| Parameter | Values |
|---|---|
| Encoder | BERT-Base |
| Context length | {256, 512} |
| Training episodes | 22,500 |
| Learning Rate | 2e-5 |
| Batch Size | 64 |
| Optimizer | Adam |

Table 9: Overview training parameters

## A.3 Model training

The model parameters are shown in Table 9. We adjust the context length according to the dataset and model. All models were trained on a NVIDIA RTX A6000 48 GB GPU. Due to memory limitations when training the model variants, we train BERT-Pair models with half-precision.

## A.4 Prompt tuning

We apply a heuristic approach in creating the template $\mathcal{T}$ and vocabulary mapping $\mathcal{R} \mapsto \mathcal{V}_{\mathcal{M}}$ used in the BERT-Prompt model. The template was selected by the authors and was applied consistently across datasets, as follows: "[TEXT]. [E1] is [MASK] by [E2].". Similarly, the tokens mapped to the target relations for each benchmarked dataset are selected by the authors. Prompt engineering, including automated search procedures for optimal templates and vocabulary mappings, remains an active research area. Acknowledging that heuristic approaches are suboptimal, we encourage further research to improve the prompt tuning approach. Our template and mapping files are available at https://github.com/pnborchert/CORE.

## A.5 Annotation Interface

The annotation platform provided clear instructions and mandatory tutorials to guide the annotators throughout the annotation process. These instructions included detailed explanations of the annotation task and interface, which was designed to be intuitive and efficient. Each component of the interface was explained in detail to ensure that annotators were able to use it effectively. The following screenshots illustrate the interface provided to the annotators. Figures 5 and 6 displays the relation annotation task, which included entity types, relation directions, and relevant context. Figure 7 shows the entity position annotation step, which verified the relation annotation completed in the previous step. Both of these steps were completed by different annotators.

# CoRE - Company Relation Extraction

## Instructions

Your task is to identify and determine the contextual relation between two entities, i.e. "Entity 1" and "Entity 2". An entity is defined as a pre-defined category including companies, products/services, brands, institutions, stock exchanges or industries. Therefore you are given a text snippet extracted from the Wikipedia page of an entity which we consider as "Entity 1". In the corresponding text, a different entity ("Entity 2") is mentioned.

The task consists of the following 5 steps:

1. Read the text snippet carefully.
2. Identify "Entity 2" in the text and verify or correct the name in the corresponding input box.
3. Select the entity type for "Entity 1" and "Entity 2".
4. Select the relation type between the two entities from the dropdown list.
5. Untick all sentences that do not contain information relevant to determine the relation between the entities.

Examples

| Relation Annotation | Entity Position Annotation |
|---|---|
| Your task is to identify and determine the contextual relation between two entities, i.e. "Entity 1" and "Entity 2".

Tutorial  Start | Your task is to verify the annotated relation and identify "Entity 1" in the context.

Tutorial |

Figure 4: Annotation Interface: Instructions.

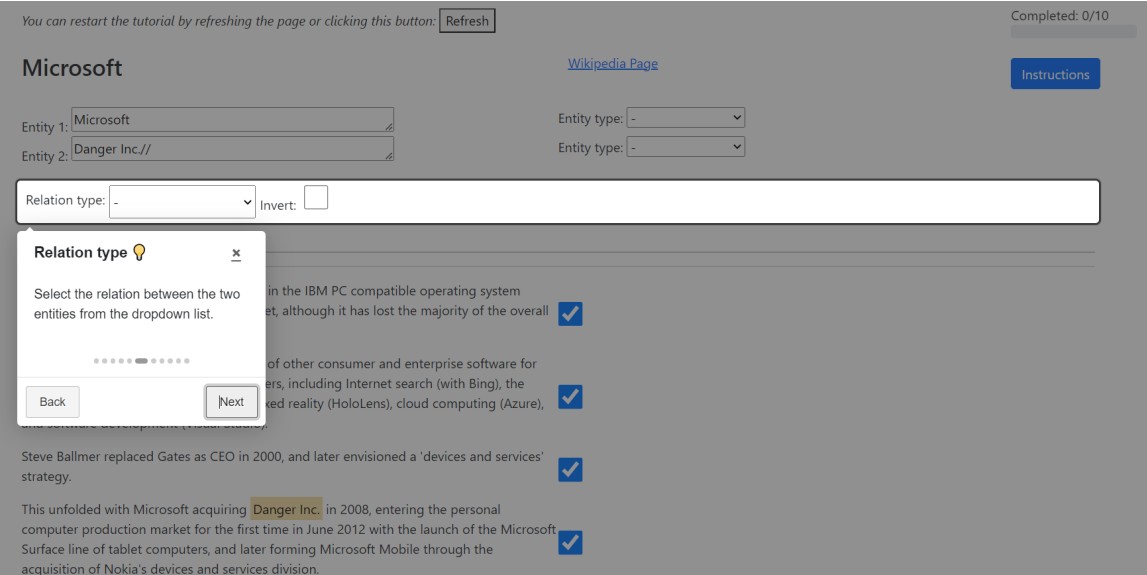

Figure 5: Annotation Interface: Relation type annotation.

# Microsoft

[Wikipedia Page](#)

Entity 1: Microsoft

Entity type: Company

Entity 2: Danger Inc.

Entity type: Company

Relation type: acquired_by    Invert: ☑

---

**Danger Inc.**  *acquired_by*  **Microsoft**

---

As of 2015, Microsoft is market-dominant in the IBM PC compatible operating system market and the office software suite market, although it has lost the majority of the overall operating system market to Android. ☐

The company also produces a wide range of other consumer and enterprise software for desktops, laptops, tabs, gadgets, and servers, including Internet search (with Bing), the digital services market (through MSN), mixed reality (HoloLens), cloud computing (Azure), and software development (Visual Studio). ☐

Steve Ballmer replaced Gates as CEO in 2000, and later envisioned a 'devices and services' strategy. ☐

This unfolded with Microsoft acquiring Danger Inc. in 2008, entering the personal computer production market for the first time in June 2012 with the launch of the Microsoft Surface line of tablet computers, and later forming Microsoft Mobile through the ☑

Figure 6: Annotation Interface: Relation type annotation.

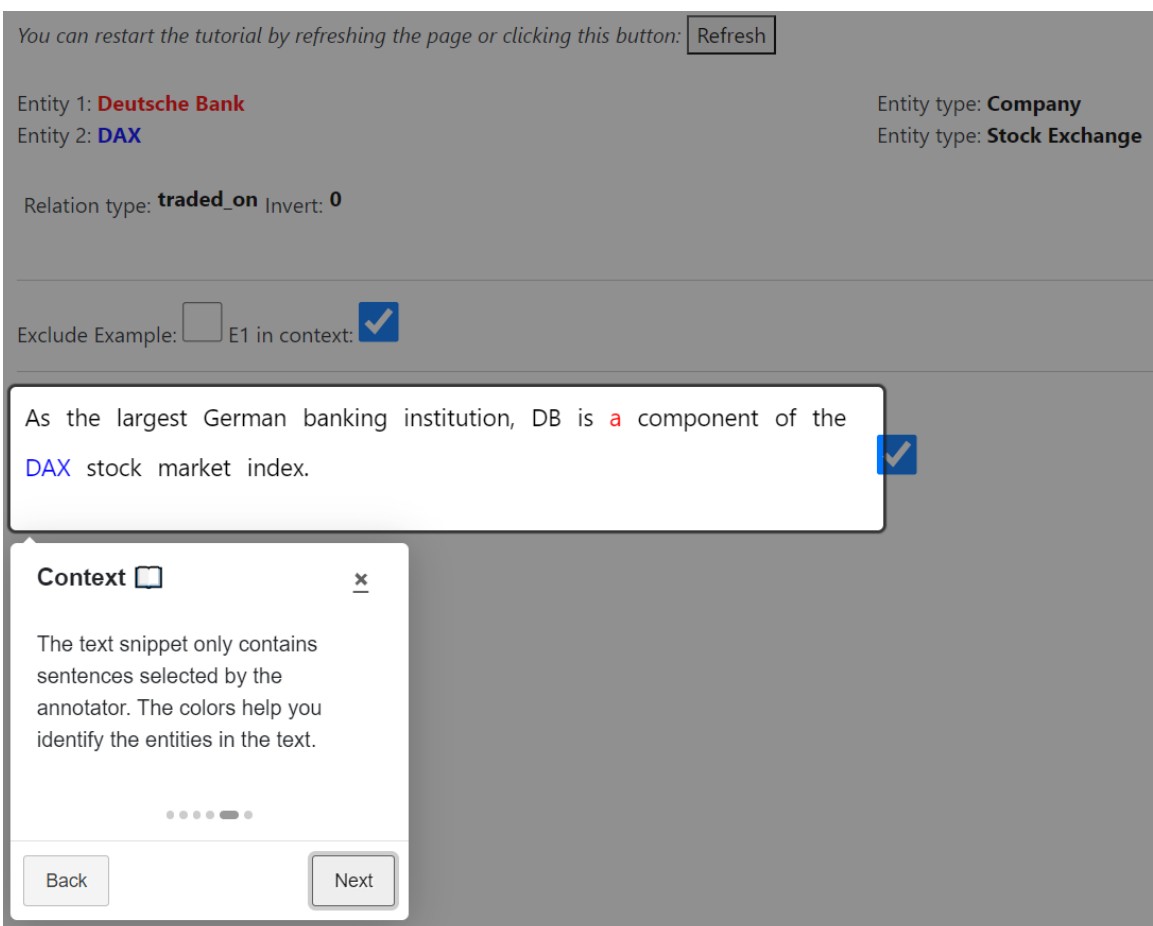

Figure 7: Annotation Interface: Position annotation and validation.