# OpenReview forum: "CORE: A Few-Shot Company Relation Classification Dataset for Robust Domain Adaptation."
_EMNLP/2023/Conference — EMNLP 2023 Main_

### Official Review · Reviewer_kwUM · 2023-08-01

**Soundness:** 3

**Excitement:**

3: Ambivalent: It has merits (e.g., it reports state-of-the-art results, the idea is nice), but there are key weaknesses (e.g., it describes incremental work), and it can significantly benefit from another round of revision. However, I won't object to accepting it if my co-reviewers champion it.

**Paper Topic And Main Contributions:**

The paper introduces CORE, a novel and high-quality dataset for few-shot relation classification (RC), and demonstrates the challenges faced by state-of-the-art models when dealing with it, while highlighting the remarkable performance of models trained on the dataset itself. The study also highlights the critical role of high-quality datasets in enabling robust domain adaptation for few-shot RC. The paper provides a benchmark evaluation of several models and approaches for few-shot RC on CORE, and conducts a qualitative analysis to identify error patterns in the domain adaptation setting.

**Reasons To Accept:**

The introduction of a new dataset is an important contribution to the field. In particular, a dataset containing containing the NOTA relation label is particularly valuable. Additionally, the fact that relations in the chosen domain would be hard to distinguish solely basing the reasoning on the entity types is a valuable characteristic of a dataset for RC.

**Reasons To Reject:**

As mentioned by the authors in the limitations sections, the fact that the dataset is based solely on Wikipedia data limits the variety of styles used in different texts in the data.
Since the authors are presenting a dataset for the few-shot setting, a dataset with a larger number of distinct relation labels would have been more useful to evaluate models on a more diverse range of classes.



**Reproducibility:**

5: Could easily reproduce the results.

**Reviewer Confidence:**

3: Pretty sure, but there's a chance I missed something. Although I have a good feel for this area in general, I did not carefully check the paper's details, e.g., the math, experimental design, or novelty.

---

> ### Author Rebuttal · Authors · 2023-08-28
>
> Thank you for your feedback, and we appreciate your recognition of the value that CORE brings to the field of relation classification.
>
> > As mentioned by the authors in the limitations sections, the fact that the dataset is based solely on Wikipedia data limits the variety of styles used in different texts in the data.
>
> We agree that including various data sources increases linguistic variety and provides a more realistic evaluation benchmark. It is important to emphasize that **expanding the dataset while maintaining the high-quality data extraction standards we uphold comes with significant costs**. Moreover, including samples from different data sources requires a careful evaluation to **minimize the introduction of linguistic clues** inherent to the data sources.
>
> We are committed to publishing our dataset under open-source licenses that enable other researchers to modify, copy and share results based on our work. Even though it comes with limitations, it's worth noting that the data extracted from Wikipedia encompasses a broad spectrum of business-related textual content curated from various primary sources (https://en.wikipedia.org/wiki/Wikipedia:Reliable_sources).
>
> > Since the authors are presenting a dataset for the few-shot setting, a dataset with a larger number of distinct relation labels would have been more useful to evaluate models on a more diverse range of classes.
>
> We acknowledge the assessment that contributions of this work and its practical applications are limited to low-resource settings, as mentioned in line 637. We carefully curated the set of relations considered in CORE to obtain a dataset that contains relations that are diverse, challenging and relevant in the business domain. Therefore, **increasing the number of relation types may inadvertently dilute the inherent complexity of the relation classification task**. The inclusion of new relation types can introduce entity type-related and linguistic clues that potentially simplify the task. An apt example would be the "location_headquarter" relation type, necessitating specific entity types such as "Company" and "Geolocation". As previously mentioned, significant costs are associated with expanding the dataset, particularly when maintaining the high-quality data extraction procedure.

---

### Official Review · Reviewer_J5cv · 2023-08-04

**Soundness:** 3

**Excitement:**

3: Ambivalent: It has merits (e.g., it reports state-of-the-art results, the idea is nice), but there are key weaknesses (e.g., it describes incremental work), and it can significantly benefit from another round of revision. However, I won't object to accepting it if my co-reviewers champion it.

**Paper Topic And Main Contributions:**

This paper proposes a few-shot relation classification (RC) dataset, CORE, that focuses on company relations and business entities. The dataset includes 4708 instances of 12 relation types extracted from various company Wikipedia pages. The dataset is targeted for robust domain adaptation. The paper shows the existing few-shot RC models struggle to generalize well to the CORE dataset as they often rely on shortcuts. However, models trained on CORE dataset display better out-of-domain performance. The paper concludes the significance of high-quality datasets such as CORE to improve model adaptation to out-of-domain data.

**Reasons To Accept:**

1. Proposes a new dataset for few-shot relation classification in company domain. The dataset requires more challenging entity types and a deeper understanding of the context compared to existing few-shot relation classification datasets.
2. Shows the dataset is challenging for out-of-domain generalization.
3. Provides insights into how diverse and challenging datasets such as CORE are required for robust domain adaptation. Encourages future research to consider these factors when generating datasets to advance research on robust few-shot RC models.


**Reasons To Reject:**

1. The scope of the paper seems a bit narrow. The dataset focuses only on company relations, but makes strong claims about generalization of RC models. Perhaps that claim needs to be supported with datasets on a few other domains.
2. The dataset is limited to company relations from Wikipedia pages. This may limit the diversity of the dataset. It may also not be representative of real-world data.
3. The dataset contains only a limited number of relations and examples, which may limit its usability in different scenarios.
4. The few-shot RC models considered in the paper are not state-of-the-art models (e.g. https://aclanthology.org/2022.coling-1.205.pdf, https://ieeexplore.ieee.org/abstract/document/10032649/). How does the performance compare to relation extraction/generation models in few-shot settings.

**Reproducibility:**

3: Could reproduce the results with some difficulty. The settings of parameters are underspecified or subjectively determined; the training/evaluation data are not widely available.

**Reviewer Confidence:**

3: Pretty sure, but there's a chance I missed something. Although I have a good feel for this area in general, I did not carefully check the paper's details, e.g., the math, experimental design, or novelty.

---

> ### Author Rebuttal · Authors · 2023-08-28
>
> We appreciate your constructive evaluation of our paper and agree with your assessment that diverse and challenging datasets such as CORE are required for robust domain adaptation. We incorporated your feedback and suggestions to strengthen the coherence and significance of our work.
>
> > The scope of the paper seems a bit narrow. The dataset focuses only on company relations, but makes strong claims about generalization of RC models. Perhaps that claim needs to be supported with datasets on a few other domains.
>
> In response to your suggestion, we **extended the benchmark datasets to include PubMed** (Gao et al. 2019), which contains **relations extracted from biomedical literature**. In the table below, we compare the out-of-domain performance of models trained on CORE, FewRel and TACRED on the PubMed dataset. We have included the evaluation results (Micro F1 / Macro-F1) for the best-performing models, namely BERT-Prompt and HCRP (Han et al., 2021). The results in the table below are averaged across three random seeds.
>
> | Domain | Model       | 5-1           | 5-5           |
> | ------ | ----------- | ------------- | ------------- |
> | CORE   | BERT-Prompt | 24.68 / 25.18 | 35.85 / 41.09 |
> |        | HCRP        | 29.74 / **52.35** | 34.96 / **55.66** |
> | FewRel | BERT-Prompt | 25.88 / 41.81 | 28.27 / 41.56 |
> |        | HCRP        | **32.41** / 40.21 | **37.99** / 36.50 |
> | TACRED | BERT-Prompt | 29.36 / 36.00 | 34.60 / 35.14 |
> |        | HCRP        | 32.00 / 45.13 | 37.14 / 48.63 |
>
> We summarize the key findings in the bullet points below:
> - The HCRP model displays the best average out-of-domain performance for all datasets.
> - In the following we evaluate the average performance across the micro-F1 and macro-F1 metrics. In the 5-1 setting, the HCRP model trained on **CORE outperforms HCRP models trained on FewRel and TACRED by 4.75% and 2.48%**, respectively.
> - In the 5-5 setting, the HCRP model trained on **CORE outperforms HCRP models trained on FewRel and TACRED by 8.06% and 2.42%**, respectively.
>
> Our manual error analysis indicates that models trained on FewRel and TACRED rely on shortcuts, such as inferring the relation type based on entity types or verbs closely related to the target relation ("sold" or "bought" for acquired by relation, "merge" for merged with relation). Hence, we conclude that the **narrow domain focus of CORE contributes to creating a challenging benchmark for few-shot domain-adaptation models** and is a valuable contribution to the field.
>
> Thank you for highlighting your concern about the assertion of strong generalization claims.
> We revised the manuscript to clarify that the out-of-domain generalization concerns the dataset included in our analysis. We include the adjustments along with the full evaluation results on the PubMed dataset in the camera-ready version of the paper.
>
> > The dataset is limited to company relations from Wikipedia pages. This may limit the diversity of the dataset. It may also not be representative of real-world data.
>
> We agree that including various data sources increases linguistic variety and provides a more realistic evaluation benchmark, as mentioned in line 629. However, it's important to emphasize that **expanding the dataset while maintaining the high-quality data extraction standards we uphold comes with significant costs**.
>
> We are committed to publish our dataset under open-source licenses that enable other researchers to modify, copy and publish results based on our work. Even though it comes with limitations, it's worth noting that the **data extracted from Wikipedia encompasses a broad spectrum of business-related textual content curated from various primary sources** (https://en.wikipedia.org/wiki/Wikipedia:Reliable_sources).
>
> Furthermore, we adhere to experimental setups that reflect real-world data (Sabo et al., 2021), and include various entity types, such as named entities, nouns and pronouns. Additionally, the relation distribution in CORE is not artificially balanced.
>
> > The few-shot RC models considered in the paper are not state-of-the-art models (e.g. https://aclanthology.org/2022.coling-1.205.pdf, https://ieeexplore.ieee.org/abstract/document/10032649/). How does the performance compare to relation extraction/generation models in few-shot settings.
>
> Thank you for this comment, we **included the HCRP model (Han et al. 2021) in our benchmark**. The updated results are presented below, reflecting both in-domain and out-of-domain performance across different datasets. **These outcomes support our earlier conclusions**: models trained on other domains struggle to adapt to CORE, while models trained on CORE showcase superior out-of-domain performance. This conclusion holds even when considering only the HCRP model.
>
> To facilitate a concise model comparison, we have included the best-performing models alongside HCRP in the tables. All evaluation results are averaged for ten sets of training and test relations for each dataset. The table below contains the evaluation results (Micro F1 / Macro-F1) for models evaluated on **CORE**:
>
> | Domain    | Model       | 5-1           | 5-5           |
> | --------- | ----------- | ------------- | ------------- |
> | In-domain | BERT-Prompt | **53.27** / **67.30** | **74.16** / **88.07** |
> |           | HCRP        | 34.82 / 38.69 | 47.40 / 50.15 |
> | FewRel    | BERT-Prompt | 25.01 / 36.35 | 33.26 / 35.70 |
> |           | HCRP        | 23.54 / 37.49 | 25.41 / 41.67 |
> | TACRED    | BERT-Prompt | 29.61 / 44.12 | 37.88 / 42.60 |
> |           | HCRP        | 27.23 / 32.85 | 41.91 / 41.61 |
>
> The table below contains the evaluation results (Micro F1 / Macro-F1) for models evaluated on **FewRel**:
>
> | Domain    | Model       | 5-1           | 5-5           |
> | --------- | ----------- | ------------- | ------------- |
> | In-domain | BERT-Prompt | **56.02** / **71.71** | 59.80 / **86.81** |
> |           | HCRP        | 36.33 / 51.07 | 52.09 / 63.11 |
> | CORE      | BERT-EM     | 45.08 / 48.01 | 51.82 / 39.05 |
> |           | HCRP        | 51.56 / 57.55 | 61.24 / 67.96 |
> | TACRED    | BERT-Prompt | 48.18 / 54.80 | 53.82 / 50.50 |
> |           | HCRP        | 49.15 / 53.89 | **61.99** / 63.33 |
>
> The table below contains the evaluation results (Micro F1 / Macro-F1) for models evaluated on **TACRED**:
>
> | Domain    | Model       | 5-1           | 5-5           |
> | --------- | ----------- | ------------- | ------------- |
> | In-domain | BERT-Prompt | 35.70 / **63.00** | **67.81** / **74.02** |
> |           | HCRP        | 29.58 / 26.43 | 57.28 / 35.61 |
> | CORE      | BERT-Prompt | **36.08** / 45.07 | 65.46 / 35.53 |
> |           | HCRP        | 34.04 / 37.54 | 47.20 / 44.76 |
> | FewRel    | BERT-Prompt | 18.71 / 49.10 | 38.21 / 45.67 |
> |           | HCRP        | 13.72 / 33.76 | 33.17 / 40.03 |
>
> We summarize the key findings in the bullet points below:
> - The in-domain BERT-Prompt model consistently demonstrates superior performance over the in-domain HCRP models for all N-Way K-Shot settings.
> - Regarding the domain adaptation to CORE, the differences between BERT-Prompt and HCRP are not as pronounced, with HCRP frequently outperforming BERT-Prompt on individual metrics. However, models trained on CORE still substantially outperform out-of-domain models.
> - For the domain-adaptation to FewRel, the HCRP models trained on CORE and TACRED are the best performing models in their category. Notably, the HCRP model trained on CORE outperforms the HCRP model trained on TACRED in three out of the four evaluation metrics.
> - For the domain adaptation to TACRED, the HCRP models do not outperform the BERT-Prompt models.
>
> We agree that the paper benefits from the addition of these results in the camera-ready version.
>
> > The dataset contains only a limited number of relations and examples, which may limit its usability in different scenarios.
>
> We acknowledge the assessment that the contributions of this work and its practical applications are limited to low-resource settings, as mentioned in line 637. We believe that few-shot evaluation scenarios and domain-adaptation are valuable to the field of information extraction.
>
> We carefully curated the set of relations in CORE. It's important to emphasize that **increasing the number of relation types may inadvertently dilute the inherent complexity of the relation classification task. The inclusion of new relation types can introduce entity type-related and linguistic clues that potentially simplify the task**. An apt example would be the "location_headquarter" relation type, necessitating specific entity types such as "Company" and "Geolocation". As previously mentioned, significant costs are associated with expanding the dataset, particularly when maintaining the high-quality data extraction procedure.
>
> References:
> - Tianyu Gao, Xu Han, Hao Zhu, Zhiyuan Liu, Peng Li, Maosong Sun, and Jie Zhou. 2019. FewRel 2.0: Towards More Challenging Few-Shot Relation Classification. In Proceedings of the 2019 Conference on Empirical Methods in Natural Language Processing and the 9th International Joint Conference on Natural Language Processing (EMNLP-IJCNLP), pages 6250–6255, Hong Kong, China. Association for Computational Linguistics.
> - Ofer Sabo, Yanai Elazar, Yoav Goldberg, and Ido Dagan. 2021. Revisiting Few-shot Relation Classification: Evaluation Data and Classification Schemes. Transactions of the Association for Computational Linguistics, 9:691–706.
> - Jiale Han, Bo Cheng and Wei Lu. 2021. Exploring Task Difficulty for Few-Shot Relation Extraction. In Proceedings of the 2021 Conference on Empirical Methods in Natural Language Processing, pages 2605–2616, Online and Punta Cana, Dominican Republic. Association for Computational Linguistics.

---

### Official Review · Reviewer_qrfS · 2023-08-05

**Soundness:** 3

**Excitement:**

3: Ambivalent: It has merits (e.g., it reports state-of-the-art results, the idea is nice), but there are key weaknesses (e.g., it describes incremental work), and it can significantly benefit from another round of revision. However, I won't object to accepting it if my co-reviewers champion it.

**Paper Topic And Main Contributions:**

In this paper, a new dataset for few-shot relation extraction, specifically company relations, is presented, which includes 4,708 instances and 12 relation types. The authors claim that models trained on different domains struggle to adapt to CORE, however, models trained on CORE show improved out-of-domain performance.
The dataset was built by gathering information from Wikipedia pages, using a state-of-the-art named entity recognition. Relations were annotated with a reasonable annotator agreement.
Although it presented the differences between CORE and FewRel, maybe an explanation is needed why this work was started while FewRel already existed. There is overlapping, maybe a combination of the two datasets would be better?
The in-domain results presented in shows that the proposed model surpassed other approaches in few-shot settings while out-of-domain evaluation reveals models trained on CORE outperform the in-domain models trained on FewRel.
The work is interesting and the dataset could be a valuable resource for few-shot domain-adaptation in relation classification.

**Questions For The Authors:**

Although it presented the differences between CORE and FewRel, maybe an explanation is needed why this work was started while FewRel already existed. There is overlapping, maybe a combination of the two datasets would be better?
The in-domain results presented in shows that the proposed model surpassed other approaches in few-shot settings while out-of-domain evaluation reveals models trained on CORE outperform the in-domain models trained on FewRel.
The work is interesting and the dataset could be a valuable resource for few-shot domain-adaptation in relation classification.

**Reasons To Accept:**

Although it presented the differences between CORE and FewRel, maybe an explanation is needed why this work was started while FewRel already existed. There is overlapping, maybe a combination of the two datasets would be better?
The in-domain results presented in shows that the proposed model surpassed other approaches in few-shot settings while out-of-domain evaluation reveals models trained on CORE outperform the in-domain models trained on FewRel.
The work is interesting and the dataset could be a valuable resource for few-shot domain-adaptation in relation classification.

**Reasons To Reject:**

Although it presented the differences between CORE and FewRel, maybe an explanation is needed why this work was started while FewRel already existed. There is overlapping, maybe a combination of the two datasets would be better?
The in-domain results presented in shows that the proposed model surpassed other approaches in few-shot settings while out-of-domain evaluation reveals models trained on CORE outperform the in-domain models trained on FewRel.
The work is interesting and the dataset could be a valuable resource for few-shot domain-adaptation in relation classification.

**Reproducibility:**

4: Could mostly reproduce the results, but there may be some variation because of sample variance or minor variations in their interpretation of the protocol or method.

**Reviewer Confidence:**

4: Quite sure. I tried to check the important points carefully. It's unlikely, though conceivable, that I missed something that should affect my ratings.

---

> ### Author Rebuttal · Authors · 2023-08-28
>
> Thank you for reviewing our manuscript. We appreciate your feedback and suggestions for improving the manuscript.
>
> > Although it presented the differences between CORE and FewRel, maybe an explanation is needed why this work was started while FewRel already existed. There is overlapping, maybe a combination of the two datasets would be better?
>
> We acknowledge an overlap between CORE and FewRel and aim to clarify the key differentiating factors that uniquely positions CORE among other relation classification datasets:
>
> - A key distinction lies in **CORE's focus on challenging entity types that require contextual understanding** and cannot be derived based on entity type information. As evidenced by our experiments, this enhances the out-of-domain performance of models trained on CORE. We believe that this **uniquely positions the scope of CORE among other relation classification datasets**.
> - The datasets fundamentally differ in their data collection and annotation methodologies, even though both datasets are based on Wikipedia. CORE is human-annotated and therefore contains **subject-object-relation triples that are not present in existing knowledge bases**. In contrast, FewRel is distantly-annotated employing external knowledge bases with subsequent human-validation.
> - We further emphasize the relation type distribution reflecting the original data source and presence of an **annotated NOTA (none-of-the-above) category in CORE**, which is essential for the application of the dataset in end-to-end relation extraction models.
>
> This results in a dataset that is substantially different in terms of content and scope. Our experimental results showcase that domain-adaptation models trained on CORE perform better in comparison to other benchmarked datasets. For applications beyond the low-resource setting evaluated in this work, we acknowledge that training on multiple datasets can be beneficial, as suggested by the reviewer.
>
> > The work is interesting and the dataset could be a valuable resource for few-shot domain-adaptation in relation classification.
>
> We appreciate your recognition of the significance of our research and the value of the CORE dataset in the field of relation extraction. We firmly believe that the CORE dataset brings a unique contribution to this domain.

---

### Meta-Review · Area_Chair_3ywq · 2023-09-21

**Recommendation:** 3

**Metareview:**

This paper proposes a few-shot relation classification dataset called CORE, focusing on company relations and business entities. The paper shows that the dataset is challenging for robust domain adaption, but the scope of the company scenario is a bit narrow.

---

### Decision · Program_Chairs · 2023-10-07

**Decision:**

Accept-Main

**Comment:**

This paper proposes a few-shot relation classification dataset called CORE, focusing on company relations and business entities. The paper shows that the dataset is challenging for robust domain adaption, but the scope of the company scenario is a bit narrow.